# Investigating the Attitude of Domestic Water Use in Urban and Rural Households in South Africa

Prince Obinna Njoku [1,*], Olatunde Samod Durowoju [2], Solomon Eghosa Uhunamure [3] and Rachel Makungo [2]

1   Department of Geography and Environmental Sciences, Faculty of Science, Engineering and Agriculture, University of Venda, P.O. Box X5050, Thohoyandou 0950, South Africa
2   Department of Earth Sciences, Faculty of Science, Engineering and Agriculture, University of Venda, P.O. Box X5050, Thohoyandou 0950, South Africa; Olatunde.Durowoju@univen.ac.za (O.S.D.); Rachel.makungo@univen.ac.za (R.M.)
3   Department of Environmental and Occupational Studies, Faculty of Applied Sciences, Cape Peninsula University of Technology, Cape Town 8000, South Africa; uhunamures@cput.ac.za
*   Correspondence: pnjoku26@yahoo.com

**Abstract:** South Africa is a semi-arid, water-stressed country. Adequate measures should be put in place to prevent water wastage. This paper aims to assess domestic water wastage and determine the proper attitude towards household water management in rural and urban communities in South Africa. This study was conceptualised in two stages. Firstly, critical observations were used to examine the attitude of households towards water usage in both urban and rural communities (Durban and Thohoyandou, respectively). Secondly, structured questionnaires and interviews were used to identify the factors that influenced the participants' attitudes towards domestic water usage. This study concludes that, irrespective of the literacy level, accessibility to limited water supply, information available through advertisements about water scarcity, and better water management in an urban community, the rural community has a better attitude towards domestic water usage and water management. The result (83.3%) also indicated that the rural community strongly agreed to be water savers in their homes. However, in the urban community, the results from the participants were somewhat evenly distributed; the participants strongly agreed and disagreed at 36.2% and 32.2%, respectively. Other results of the study also showed that variables such as family upbringing, inaccessibility of domestic water, and advertisement play a major role in influencing the attitude of the rural community to water usage. These variables were statistically significant at $p < 0.001$. However, the immediate environment was shown to be not statistically significant at $p < 0.911$. Based on the study results, it is recommended that households should be encouraged to generate greywater collection systems to reduce water use and improve water reuse. The government could introduce a rationed allocation (shedding) of domestic water in urban communities to draw attention to the prevalence of water scarcity in the nation.

**Keywords:** attitude; domestic water use; South Africa; water management; water conservation

## 1. Introduction

Freshwater is an essential natural resource and is vital for sustaining life and supporting the development of ecosystems and recreational sources. Therefore, water should be used sustainably. The ideology of the sustainability of freshwater resources is using the resource as a means to meet the needs of the present generation without compromising the needs of the future generation [1]. However, while water is a renewable resource, it is finite; thus, there are certain consequences of meeting the needs of the present and future generations since water demand often far exceeds its availability [2]. Globally, the demand for domestic water has continued to increase due to the increase in population growth, urban migration, food demand, standard of living, and global wealth. There have been

several solutions to this issue, for instance, on the macro scale through major desalination of wastewater treatment plants, construction of more dams, tapping of underground water supplies, and recycling industrial wastewater, and on the micro or domestic scale, through the installation of water tanks, recycling household greywater, and other domestic practices. There are also several initiatives to mitigate the consumption of domestic water by using efficient fittings within homes and by encouraging changes in gardening practices. These practices have had little or no impact on the widespread shift in water use attitude and behaviour [3].

The increase in global water use aggravates the water scarcity conditions, especially in arid nations such as South Africa, where precipitation is lower, which limits available surface water and affects the attitudes of individuals towards the use of available water resources. As the global household demand for water usage continues to rise, a question arises concerning the scarcity and the attitudes of households towards water use. South Africa is a water-scarce country and the 30th driest in the world and is already feeling the pressure of the prediction that the country's water demand will outweigh its availability by 2030 [4]. It is therefore important not to misuse the available domestic water. The focus on domestic water management and ways of reducing water wastage should be reinforced from the household level. Inadequate rainfall, climate change, rapidly growing population, a growing economy, and other issues stem from the unavailability of water resources [5]. This has increased the rationing of domestic water in several places, and it is a more common practice in rural areas. The rationing of domestic water is likely to become a fact of life as time goes on, affecting the urban centres likewise. Moreover, water conservation measures must be implemented in all aspects of life as a matter of urgency [6]. Government and relevant stakeholders can facilitate sustainable water management practices in so many ways by supply augmentation (water recycling and desalination) and demand management practices aimed at reducing consumption. The change in the attitude and behaviour of individuals towards the use of water is also a very vital and essential tool for water management [1–3].

South Africa is currently exploring 98% of its available freshwater resources, and about 45% of its domestic water cannot be accounted for [7]. An average South African household uses about 250 L/day of water, which is more than the average amount of water (173 L/day) recommended globally [8]. Despite South Africa's severe drought with most metropolitan areas instituting water restrictions, many South Africans still consume more water than the global average. The need for domestic freshwater by households includes drinking, hygiene, washing, cooking, gardening, and productive purposes such as farming, livestock, forestry, fisheries, and small-scale industries. However, only 24%, if not less, of rural households in South Africa have access to piped water [9]. Wasteful water practices continue to loom in both urban and rural communities, especially in South African households. A study by the Council for Scientific and Industrial Research (CSIR) showed that increasing access to water leads to increased water usage. This is particularly predominant in an urban household; nonetheless, this wastage is caused by non-maintenance of household infrastructure, especially in poor households [10]. Therefore, rather than focusing on increasing water supply, it is critical to focus on the behaviour and attitude of individuals towards the consumption of water, as well as on how to mitigate high water usage in both urban and rural communities, in turn reducing water demand and drastic high water use across the country. Water use and the demand side policy responses to mitigate future domestic water variability will benefit from a deeper understanding of current household water use, perception of water use, attitude towards water use, and factors influencing household water use. Such a deeper understanding will address key questions such as the following:

1. Do South Africans understand that the country is semi-arid with limited water resources?
2. When asked to reduce the amount of water being used, would people know which behavioural changes are more effective than others?

3. What makes people use water in the manner that they do, and what will motivate them to change how they use water?

4. Are people conscious of the amount of actual water used, and do their perceptions of water use correspond with actual water use?

It is on the aforementioned premise that this paper seeks to understand the behaviour and attitude towards water use in the surveyed communities. The results of the study will enable relevant stakeholders to inculcate a proper attitude towards water management for South Africans living in rural and urban communities to minimise high water usage. Recommendations will be given on how to save water. This paper will also help proffer possible solutions to save water and a positive step towards an instrumental water management strategy in South Africa.

## 2. Materials and Methods

### 2.1. Location of Study

The study was conducted in two distinct locations in South Africa: Thohoyandou (Vhembe District) in Limpopo Province and Durban (eThekwini District) in KwaZulu-Natal Province (Figure 1).

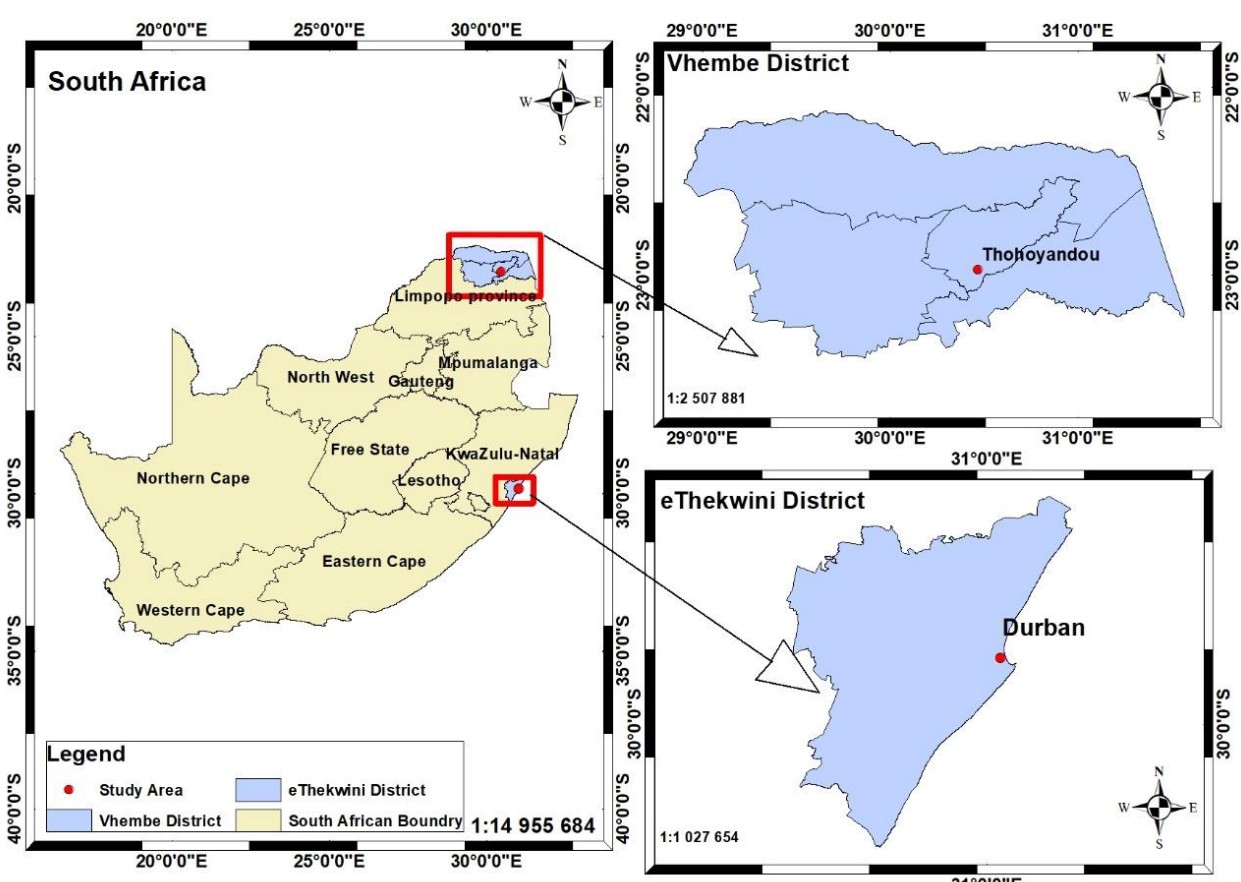

**Figure 1.** Map of the study areas.

Vhembe District is located in the northernmost part of the Limpopo Province and is made up of four local municipalities: Thulamela, Makhado, Musina and Collins Chabane. It has a population size of about 1,294,722 people and 335,676 households. The district borders the Kruger National Park (East), Capricorn District (Southwest), Botswana (Northeast) and Zimbabwe (North). The main languages spoken are Tshivenda (69%) and Xitsonga (27%) [11]. It is mostly a rural-based district, and households are headed by females while the males migrate to cities for work [12]. According to Vhembe District Municipality [11], approximately 128,372 households in the district have access to water

under the Reconstruction and Development Program (RDP), which aims to improve the standard of living for South Africans. Furthermore, 22,835 households have access to water via boreholes, springs, rivers, streams, dams, rainwater, and water vendors. The district has about 11 dams, which are the major sources of domestic water. The district strives to provide free basic water to low-income households. The province is a typical developing area and is regarded as one of the poorest regions of South Africa, with an immense gap between poor and rich residents, especially in the rural areas. Most of the households in the rural areas, which comprise much of the population of the province, depend on pension grants, government grants, and remittances from family members who migrated to other provinces to work. The household wealth is relatively lower compared to other municipalities in South Africa [13]. Thohoyandou is the largest town in the Vhembe district, and it is mainly made up of rural townships and is further surrounded by several rural villages situated on the outskirts of the built-up community. The underdeveloped rural area experiences periodic droughts and environmental degeneration due to poor water supply and infrastructural development. Despite recent improvements, rural people still use water directly from streams for household purposes, which include washing of laundry, car washing, and bathing directly in the river. The area consists of approximately Black African 95.5%, Coloured 0.2%, Indian/Asian 4.1%, White 0.2% and other races 0.1% [14].

The Durban community is a coastal urban city located in eThekwini Municipality in the eastern part of South Africa. The city is the third most popular city in South Africa and the largest city in KwaZulu-Natal Province. According to the last census Statistics South Africa [15], Durban has a population of approximately 3.44 million people, with a race makeup of 51% Black Africans, 8.6% coloured, 24% Indians, 15.3% Whites, and 0.9% other races. Durban Metropolitan City covers an area of 1370 km$^2$ and stretches 72 km along the Indian Ocean and 52 km inland. The Durban Metro Water Service (DMWS) is in charge of water supply, sanitation and solid waste and currently serves 360,000 households with metered water connections. However, approximately 43% of these households lack household connections. Stolen water is also very common in Durban; approximately 35% of the city water is stolen or given out through illegal means. The households, therefore, depend on stand posts, many of which are inherited by the DMWS from the previous administration. Meanwhile, there are approximately 10,000 to 20,000 illegal water connections in the city's piped system [16].

### 2.2. Research Approach

This study employed a quantitative and qualitative approach which was conducted in two stages.

Stage 1: A reconnaissance survey of two weeks (4 July to 18 July 2019) was conducted to identify and understand the demography of the participants to be included in the study. During the reconnaissance survey, two researchers went to both communities to survey the area. Upon the ground survey, a simple random sampling technique was best fit to identify the participants for this study [17]. Due to the similar water management and challenges experienced in both communities, the participants were randomly selected from both communities. Ten households were selected in the rural community and ten households in the urban community that were relevant observatory participants for the study, as shown in Table 1. Durban is an urban community, while Thohoyandou is a rural community. Furthermore, data collection was done through the observatory technique and structured interview; this is to understand which of the two communities (urban and rural) have better attitudes towards water management [18].

Stage 2: A reconnaissance survey for one week (22–29 July 2019) was conducted to identify possible participants. Reconnaissance survey is the site investigation and is carried out at the preliminary stage before other stages are begun. Reconnaissance involves a field trip to the site where further investigation is to be carried out. It gives details of landforms and other structures that are above ground, which may form an obstacle for an installation. Raosoft software was manufactured by the Raosoft, Inc. group (Seattle, WA, USA) was used

to arrive at a sample size [19]. The statistical software is used for calculating sample size, which comprises a database management system of great strength and reliability that also communicates with other proprietary formats. The Raosoft database is an extremely robust, proven system with high data integrity and security. The software assumes a marginal error of 5% and a confidence level of 92% to ascertain the amount of uncertainty to be tolerated. Then the combined population size of both communities is inputted into the software.

**Table 1.** Demographic characteristics of the participants in stage 1.

| Households | Total Number in Household | | Male | | Female | | Number of Children | | Head of Household | |
|---|---|---|---|---|---|---|---|---|---|---|
| | Rural | Urban | Rural | Urban | Rural | Urban | Rural | Urban | Rural | Urban |
| 1 | 5 | 5 | 2 | 2 | 3 | 3 | 2 | 2 | Grandmother | Father |
| 2 | 8 | 3 | 3 | 1 | 5 | 2 | 5 | 2 | Grandmother | Grandfather |
| 3 | 4 | 4 | 1 | 1 | 3 | 3 | 2 | 1 | Grandfather | Father |
| 4 | 5 | 6 | 2 | 3 | 3 | 3 | 1 | 3 | Mother | Aunty |
| 5 | 3 | 2 | 2 | 1 | 1 | 1 | - | 1 | Mother | Father |
| 6 | 6 | 3 | 2 | 1 | 4 | 2 | 3 | 1 | Father | Father |
| 7 | 7 | 4 | 4 | 1 | 3 | 3 | 3 | 1 | Grandmother | Mother |
| 8 | 4 | 6 | 2 | 4 | 2 | 2 | 2 | 2 | Mother | Mother |
| 9 | 3 | 4 | 1 | 3 | 2 | 1 | 1 | 1 | Grandmother | Grandmother |
| 10 | 3 | 5 | 1 | 2 | 2 | 3 | 2 | 2 | Father | Mother |

The sample size n and margin of error E are given by

$$x = Z \ (c/100)^2 \ r \ (100 - r) \tag{1}$$

$$N = Nx/((N - 1)E^2 + x) \tag{2}$$

$$E = \sqrt{(N - n)x/n(N - 1)} \tag{3}$$

where N is the population size, r is the fraction of responses that you are interested in, and Z(c/100) is the critical value for the confidence level c [19]. After running Raosoft software for this study, a total of 307 participants were drawn for the study, which included both rural and urban communities.

However, 229 respondents participated fully, with 102 and 127 participants from rural and urban communities, respectively. A random sampling technique was used to identify the participants for the study. The participants were selected based on their geographical location; all participants currently lived in Durban or Thohoyandou. Structured questionnaires and interviews were administered in the English language, but where unavoidable, the local dialects (mostly Tshivenda and IsiZulu) of the participants were used. To find out the factors that influenced their attitude towards domestic water wastage, a quantitative approach was used.

### 2.3. Data Collection

An in-depth, conscious observation process was conducted with the participants to collect data from their homes. The observatory process was conducted for 30 days (4 August to 3 September 2019). Daily, the participants were asked to identify which water variables were applicable in their households during the data collection process. The water variables guide used for stage 1 includes water reuses, water-saving attitude, water availability, water management and frequency of water use.

In stage 2, five students were trained to facilitate the administration of the questionnaires. The variables guide that was used for this stage includes family upbringing,

immediate environment, education, advertisements and inaccessibility/scarcity of water. To ensure quality assurance and quality control regarding the reliability and validity of the data to be collected, the questionnaires were pretested before the main survey, with adjustments and corrections made based on the received responses to improve clarity.

*2.4. Data Analysis*

Thematic and content analyses were used to analyse the qualitative data obtained [20]. Moreover, the IBM Statistical Package for Social Science (SPSS) for Windows, version 26 (IBM Corp., Armonk, NY, USA) was used to analyse the quantitative data obtained. The data derived from the participants were validated by checking if all questionnaires were correctly filled and checked for errors and by ensuring that the data were enough to generalise the findings of the study. Descriptive statistics analysis using percentage and frequency was performed to describe and conceptualise the data. A normality test was conducted to understand the type of statistical analysis that best fits the data derived. Subsequently, after considering the assumptions for analysing the variables from two independent groups, an independent sample *t*-test analysis was performed to compare the difference in means derived from the two groups. To find the correlation between the different variables, Spearman's rho correlation analysis was conducted.

*2.5. Ethical Clearance*

This study was approved by the ethical committee of the University of Venda (certificate number: SES/21/ERM/06/0306). To ascertain the avoidance of harm to the participants, a consent form was handed to them, explained to them and signed by them before data collection. The participants were guaranteed strict confidentiality and anonymity of the data they provided. The participation was voluntary, and where a household declined to participate, another household was randomly selected.

**3. Results and Discussion**

*3.1. Stage 1*

The notion of domestic water running out and the sense of water as a precious commodity was the main concern for some of the participants in the rural community. It was observed that most participants in rural households reuse water compared to the participants living in urban areas. A grandmother in the rural community showed her concerns; she said, "*Water is scarce and is very precious to us, and we never know when the taps are going to run again.*" One of the mothers also said, "*We have to save as much water as possible whenever the municipality gives us water.*" A child from the rural household noted, "*We do not waste water in our house; otherwise we will be in trouble with Mama.*" A study in Cape Town, South Africa, shows that the residential water consumption trend found that washing of clothes was perceived to be the highest water use activity compared to other water use activities in informal settlements [21]. The result of a study in rural southwest Victoria in Australia indicated that water usage coupled with water-saving devices and multiple saving behaviours using (water-saving tanks in their homes) reduced water wastage. This is unlike residents who had a high supply of water and did not have water tanks but still believed that they did not waste water [22]. With regard to washing clothes, this study explored and found that, in most of the rural households, the water used for washing clothes in their homes is either stored or reused to flush their toilet or pit system. This was because the participants in the rural community did not have easy access to water for domestic use, so they had to reuse water for different purposes. However, all participants in the urban households pour away the water used for washing clothes. One of the participants in the rural community explained that they do not have enough water supply given to them by the municipality, "*so we have to manage the water we have.*"

The urban community was more relaxed and less conscious of water wastage, although parents sometimes complained about water wastage because of compounding water tariffs and bills. A household of four living in the urban community was observed, and it was

found that the inhabitants showered at least twice a day, once in the morning and once in the evening. Moreover, while brushing their teeth, the children left the bathroom sink tap running. Washing of clothes was done with ease, with the tap running at all times. According to one of the children, "*There is always water in our house, this is why I leave the tap running while washing my teeth.*" Another participant said that the "*tap in the bathroom is broken and the water comes out with much pressure, so I have to lower the tap and let it flow slowly till I finish with my laundry.*" The used water was poured into the bathroom sink and not reused. The urban community was observed to be high water users compared to the rural community. Water leakage from broken taps, frequent showering and washing of clothes by family members, despite the rapid increase in water scarcity, were some of the observations. The urban community perceived domestic water as a right and the responsibility of the government to provide. However, the households that saved water in the urban community did that because of accumulating water bills and not because of the looming presence of water scarcity in the country. On the contrary, Gilbertson et al. [2] explained that households in Australia with a good supply of water could also conserve water by making sure that taps do not drip, having a dual flush toilet, using the washing machine only when it is full and using minimal water for cleaning.

It should be noted that the rural community had major problems with the easy access and regular availability of domestic water supply compared to the urban community. It is clear that the easy access and ready availability of domestic water highly influence both communities on water conservation. The participants living in the rural community had a better water-saving attitude than the participants living in the urban community. This is mostly a result of the high scarcity/inaccessibility of domestic water in the rural community. Meanwhile, the participants living in the urban community expressed their water culture based on the premise of continuous flow and regular availability of water. The government has maintained a high delivery standard for water supply in the urban community. However, key steps to water management and water usage were also limited in the urban community, for example, regulated tap mouth, reuse of greywater after washing, prompt fixing of leaking pipes and fines to households that flout the water restriction rules by the government.

The fear of the community running out or not receiving water was also prominent in the minds of most participants living in the rural community. It was observed that most of the participants living in urban areas have domestic water readily available compared to the participants from rural households. For example, some homes in the rural community receive water only on Friday (for one hour), Saturday (10 a.m.–12 p.m.) and Sunday (10 a.m.–12 p.m.). This makes the families very conscious about saving enough water in the event when the community taps stop running. Bathing, washing, cooking, among other activities, are carried out with all consciousness because of poor access to water. Furthermore, a household living in the rural community receives water only on Mondays. Sometimes, throughout the whole week, water is not received at all. This has made some wealthy families drill personal boreholes for easy access to water. However, the water from these boreholes is salty and unsuitable for some domestic purposes such as drinking. One of the participants in the rural community explained that as a household chore that is often directed to the females in the family, due to domestic water needs, the females have to walk long distances to get to the community borehole for clean water. They ensure that the water tanks are always filled with water at all times from the community tap. A household of six members (one grandparent, two parents and three grandchildren) living in the rural community was observed and it was discovered that the grandchildren bathed only in the morning with a small amount of water before going to school. Water is so precious to them that any form of wastage by the grandchildren will result in scolding by the parents. Grandma says, "*Water is very scarce in our community, and it takes much effort to get clean water. Sometimes I fear this water will stop running in our community.*" Overall, the rural community were seen to be water savers and more conscious about their water use than the participants from the urban community. Similarly to the above result, a study conducted in

Nepal, Kathmandu Valley, which suffers from poor water management, showed that the households in the Valley engage in five main types of water-coping behaviours, which are collecting, pumping, treating, storing and purchasing [23]. These coping strategies include walking far distances for water, construction of private wells and boreholes, rainwater harvesting and purchase of water from vendors and neighbours. The study further showed that these coping behaviours were converted into monetary value and that the poorest households incurred four times less coping cost than the top 20% households with higher earnings [23].

With minimal availability of water for bathing, washing, rinsing of clothes and cooking, life still goes on for the participants living in the rural community. Unlike the participants observed in the urban community, they have a high number of running taps in almost every room of the house and can afford to shower twice a day, rinse their clothes with two or three full buckets of water, and even have lawns and gardens to water. The urban area participants were observed to have various water source outlets in their homes such as kitchen sink taps, toilet sink taps, showers, water closets and backyard taps. If only the urban community could learn and emulate some water-saving culture from the rural community, water problems would be reduced to a minimum in the country. Some water conservation strategies include installing storage tanks, rainwater harvesting, reduced number of taps, reduced use of water in homes, and reducing the amount of water being poured away. Urban areas across the world with good water supply have shown that household water use will continue to rise as long as global urbanisation continues to increase. Better water supply drives up demand. Moglia et al.'s [24] review shows to what extent water conservation strategies have influenced water use in urban areas with references from the US, the UK, Australia and Spain. These countries sometimes experience drought/water scarcity but enjoy better water management. The study showed that rainwater harvesting is the most effective among the conservation strategies; however, rainwater harvesting comes with significant investment cost and requires ongoing maintenance and operation by the households. Meanwhile, public awareness and media campaigns were the most consistent and effective water conservation mechanism, especially during water crisis periods such as drought. Moving from fixed pricing to volumetric pricing was also a considerable strategy for impact in terms of water savings. The study also calls for a more effective water pricing whereby households that use larger amounts of water are billed higher. However, this has been criticised as unfair because the amount of water consumed is heavily influenced by the size of the household, and larger households are often associated with lower socioeconomic position. The strategies of Moglia et al. [24] are similar to the results derived from this study; yet it is argued that water conservation strategies are unique and specific to different locations due to varying unique water challenges.

### 3.2. Stage 2 Experiment

The experiment looked into certain variables such as family upbringing, advertisements, the immediate environment and inability to understand why there are discrepancies in rural and urban communities towards domestic water usage. The demographic characteristics of the participants were analysed to understand the social characteristics of the participants used in this study. Table 2 reveals the results obtained from the study.

Table 2 shows that there was no statistically significant difference between the two communities (*p*-value at 0.980) for gender. This implies that gender type in both communities (Thohoyandou and Durban) does not influence water wastage as water is essential to all genders. This finding aligns with the result obtained by Graymore and Wallis [22]. However, more female respondents were observed in both communities because of their availability and readiness to participate in the study. The statistical difference between the two communities for age was not significant (*p*-value at 0.513); participants aged 21–30 years old (44.1%) were the most dominant age group in the rural community that participated in this study. However, participants aged 16–20 years old (39.4%) were the most dominant in the urban community. Although most of the participants in the rural

community were in the age group of 21–30, the majority of them had attained their highest educational qualification at the secondary level. On the contrary, most of the urban participants attained their highest education qualification at the tertiary level. Consequently, the statistical difference between the two communities for educational attainment was statistically significant ($p < 0.001$). Moreover, most of the rural respondents have lived more than 15 years in the community; however, the urban participants have lived more than six years or above in the community. The statistical difference between the communities for how long the respondents have lived in the area was statistically significant ($p < 0.001$). This implies that the level of education plays a significant role in the water consumption in their various communities. However, the rural community participants were less educated than those in the urban community but were more prudent with water usage (more water-wise). This could be as a result of irregular supply of water, old water infrastructures, lack of proper maintenance of water facilities, among other factors.

**Table 2.** Demographic characteristics of the participants.

| | Residents Living in the Rural Community | | Residents Living in the Urban Community | | Significant |
|---|---|---|---|---|---|
| | **Frequency** | **Percentage** | **Frequency** | **Percentage** | |
| Gender | | | | | |
| Male | 44 | 43.1 | 55 | 43.3 | |
| Female | 58 | 56.9 | 72 | 56.7 | 0.980 |
| Total | 102 | 100 | 127 | 100 | |
| Age (years) | | | | | |
| 16–20 | 26 | 25.5 | 50 | 39.4 | |
| 21–30 | 45 | 44.1 | 26 | 20.5 | |
| 31–40 | 9 | 8.8 | 29 | 22.8 | |
| 41–50 | 13 | 12.7 | 14 | 11 | 0.513 |
| 51 and above | 9 | 8.8 | 8 | 6.3 | |
| Total | 102 | 100 | 127 | 100 | |
| Educational attainment | | | | | |
| No level of education | - | - | 5 | 3.9 | |
| Primary | 35 | 34.3 | 22 | 17.3 | |
| Secondary | 48 | 47.1 | 31 | 24.4 | |
| Tertiary | 18 | 17.6 | 69 | 53.3 | 0.001 |
| Did not tell | 1 | 1 | - | - | |
| Total | 102 | 100 | 127 | 100 | |
| How long have you lived in the community (years) | | | | | |
| Less than 1 | 2 | 2 | 13 | 10.2 | |
| 1–5 | 3 | 2.9 | 23 | 18.1 | |
| 6–10 | 16 | 15.7 | 25 | 19.7 | |
| 11–15 | 20 | 19.6 | 33 | 26 | 0.001 |
| Above 15 | 59 | 57.8 | 33 | 26 | |
| Did not tell | 2 | 2 | - | - | |
| Total | 102 | 100 | 127 | 100 | |

At a significant level of 0.05 ($p < 00.05$).

*3.3. Water Saver*

　　As indicated in Table 3, the participants' perceptions regarding household water-saving show a statistical significant between the two communities ($p < 0.001$), rejecting the null hypothesis. This implies that there is a significant difference in the mean of the participants' perception towards water saving among the two communities. The results of the survey in the rural community show that 83.3% strongly agreed to be water savers in their homes. This could be a result of the inaccessibility to or scarcity of water in their community. However, in the urban community, the results from the participants were somewhat evenly distributed; the participants strongly agreed and disagreed, at 36.2% and 32.2%, respectively. This may be attributed to their households receiving a constant water supply and paying some of the municipality's water charges.

**Table 3.** Perception of the participants towards domestic water usage.

| | Residents Living in the Rural Community | | Residents Living in the Urban Community | | Significant |
|---|---|---|---|---|---|
| | **Frequency** | **Percentage** | **Frequency** | **Percentage** | |
| Do you feel you and your household are water savers? | | | | | |
| Strongly agree | 85 | 83.3 | 46 | 36.2 | |
| Agree | 4 | 3.9 | 17 | 13.4 | |
| Maybe | - | - | - | - | |
| Disagree | 13 | 12.7 | 23 | 18.1 | 0.001 |
| Strongly disagree | - | - | 41 | 32.3 | |
| Total | 102 | 100 | 127 | 100 | |
| Do you think your family upbringing influences your use of water? | | | | | |
| Strongly Agree | 76 | 74.5 | 72 | 56.7 | |
| Agree | 10 | 9.8 | - | - | |
| Maybe | 3 | 2.9 | 16 | 12.3 | |
| Disagree | 13 | 12.7 | 22 | 17.3 | 0.001 |
| Strongly disagree | - | - | 17 | 13.4 | |
| Total | 102 | 100 | 127 | 100 | |
| Have you seen any form of awareness of water scarcity before? | | | | | |
| Seen | 24 | 23.5 | 97 | 76.4 | |
| Maybe | 4 | 3.9 | 3 | 2.4 | |
| Not seen | 72 | 70.6 | 27 | 21.3 | 0.001 |
| Did not tell | 2 | 2 | - | - | |
| Total | 102 | 100 | 127 | 100 | |
| Does your immediate environment influence your use of water? | | | | | |
| Strongly agree | 76 | 74.5 | 101 | 79.5 | |
| Agree | 5 | 4.9 | - | - | |
| Maybe | - | - | 1 | 1 | |
| Disagree | 18 | 17.6 | 11 | 8.7 | 0.911 |
| Strongly disagree | 3 | 2.9 | 14 | 11 | |
| Total | 102 | 100 | 127 | 100 | |

**Table 3.** *Cont.*

| | Residents Living in the Rural Community | | Residents Living in the Urban Community | | Significant |
|---|---|---|---|---|---|
| | Frequency | Percentage | Frequency | Percentage | |
| Is there water inaccessibility or scarcity in your community? | | | | | |
| Strongly agree | 85 | 83.3 | 10 | 7.9 | |
| Agree | - | - | 18 | 14.2 | |
| Maybe | - | - | - | - | |
| Disagree | 14 | 13.7 | - | - | 0.001 |
| Strongly disagree | 3 | 2.9 | 99 | 78 | |
| Total | 102 | 100 | 127 | 100 | |

At a significant level of 0.05 ($p < 0.05$).

*3.4. Family Upbringing*

Family upbringing shows how the participants are influenced by other family members (especially parents) towards the behaviour and perception of water issues. The role of parents in the family influences the behaviour and perception of other family members. Table 3 indicates that the statistical difference between the communities for family upbringing was statistically significant ($p < 0.001$). This implies that there is a significant difference in the mean of the participants' perception towards family upbringing and water use among the two communities. Consequently, most of the participants in the rural and urban communities (74.5% and 56.7%, respectively) strongly agreed that their family members were influenced by their attitude towards water use. The rural community was more influenced by family upbringing than the individuals in the urban community. This result is in line with the findings of [25], which shows that parenting and its effect on children are clear evidence that parents influence the behaviour/perception of their children. Although individuals seemed to be influenced by their immediate environment, parenting still plays a significant role in an individual's attitude towards water use.

*3.5. Immediate Environment*

Table 3 shows that there was no statistically significant difference in the immediate environment between both communities *($p < 0.911$)*. This implies that there is no significant difference in the mean of the participants' perception towards their immediate environment and water use among the two communities. Most participants from both communities strongly agreed that the immediate environment influences the way water is used. Moreover, from the stage 1 experiment, the immediate environment showed a strong influence on the attitude of individuals towards water use. From the stage 1 experiment, a household of six people living in the rural community did not receive water at all in their community and had to walk long distances to get clean water. The grandma stated how the neighbour influenced her: *"My neighbour wakes up very early in the morning to fetch clean water from the other streets. Also, my neighbour installed a big water tank to save water whenever tankers selling water come around to sell water."* Ellen and Turner [26] showed that one's immediate environment (neighbourhood) plays a vital role in the development of one's behaviour in the different stages of one's life. Moreover, Larson and Brumand [27], in their study of paradoxes in landscape management and water conservation ("Examining Neighbourhood Norms and Institutional Forces"), showed a significant relationship between water conservation and neighbourhood pressure.

Larson and Brumand [27] showed that the neighbourhood positively influences the attitude towards water use in the community. However, this study agreed with the reviewed literature and showed that in the rural community, the immediate environment strongly influences the attitude towards water use. Furthermore, similar results were obtained in the urban community. Therefore, the highly significant impact of the immediate environment

on water use contributes to the reason why the participants living in the rural community are better water savers. Moreover, the immediate environment and the availability of water play a significant role in explaining why the participants living in the urban community are heavy water users.

### 3.6. Inaccessibility/Scarcity

Scarcity/inaccessibility shows the accessibility of water in both communities. The statistical difference between the communities in inaccessibility was statistically significant at $p < 0.001$. This implies that there is a significant difference in the mean of the participants' perception towards scarcity/inaccessibility among the two communities. As indicated in Table 3, 83.3% of the rural participants strongly agreed with the issues of water scarcity in their community. However, 78% of the participants in the urban community strongly disagreed with water scarcity in their community. The participants identified that water sometimes stops running when the municipality wants to perform or undertake a water purification routine. Overall, the urban community was not ignorant of the water scarcity in the country (Table 3).

### 3.7. Advertisement

Advertisement highlights the rate of awareness/education of water scarcity in South Africa. The statistical difference between the communities in advertisement was statistically significant at $p < 0.001$. This implies that there is a significant difference in the mean of the participants' perception towards advertisement and water use among the two communities. As shown in Table 3, the participants from the urban community were more aware of water scarcity (76.4%) than the participants in the rural community (23.5%). Furthermore, scholars have indicated that the provision of information improves water conservation habits [28]. However, despite the constant advertisements on television, workshops and billboards by the government and other water stakeholders concerning water wastage and water management, the participants living in the urban community were observed to have a poor attitude towards water wastage compared to the participants living in the rural community.

Moreover, Spearman's rho correlation (Table 4) was run to determine the relationship between advertisements and water savers. This was done to understand how perceived water savers are associated with advertisements of water-related issues. The result was statistically significant ($p < 0.001$), and there was a low negative correlation between advertisement and water saving (0.306). The more the advertisements on water issues, the more likely it is that water will be saved in communities.

**Table 4.** Spearman's rho correlations between advertisements and water scarcity and water savers.

| | | | | |
|---|---|---|---|---|
| Spearman's Rho | Have You Seen or Heard Any Form of Awareness on Water Scarcity before | Correlation Coefficient | 1.000 | 0.306 ** |
| | | Sig. (2-tailed) | | 0.001 |
| | | N | 229 | 229 |
| | Water Saver | Correlation Coefficient | 0.306 ** | 1.000 |
| | | Sig. (2-tailed) | 0.001 | |
| | | N | 229 | 229 |

** Correlation is significant at the 0.05 level (two-tailed).

Sarkar et al. [29] emphasised that the introduction of environmental education would be an effective tool for water resource management. Publications of leaflets, books, posters and workshops in regional language focusing on environmental strategies will be a viable tool for water management. Nieswiadomy [30] researched price structure, conservation and education to estimate urban residential water demand. It was discovered that public education had a significant impact on reducing water wastage. However, in this study, the participants living in the rural community had a lower level of education but still had a

better attitude towards water saving and wastage than those from the urban community. Therefore, this study contradicts Nieswiadomy's study [30] by arguing that water wastage and water management cannot be entirely reduced by one's level of education, but rather by experience, the environment and other major drivers.

Moreover, Spearman's rho correlation (Table 5) was tested to determine the relationship between scarcity/inaccessibility of water and water use. The result was statistically significant ($p < 0.0001$), and there was a low positive correlation between inaccessibility of water and water use (0.342). The more the participants who have access to water, the more likely it is that less water will be saved in the communities.

**Table 5.** Spearman's rho correlations between water savers and water inaccessibility/scarcity.

|  |  |  |  |  |
|---|---|---|---|---|
| Spearman's rho | Water Saver | Correlation Coefficient | 1.000 | 0.342 ** |
|  |  | Sig. (2-tailed) |  | 0.001 |
|  |  | N | 229 | 229 |
|  | Inaccessibility/Scarcity | Correlation Coefficient | 0.342 ** | 1.000 |
|  |  | Sig. (2-tailed) | 0.001 |  |
|  |  | N | 229 | 229 |

** Correlation is significant at the 0.05 level (two-tailed).

A study by Jacobs-Mata [10] reviewed several studies investigating the interrelationship between individuals' attitudes towards water use and social-demographic factors such as income, education, political affiliation, family size, type of dwelling, water inaccessibility, advertisement and homeowners. Some of the results showed that income and water conservation have a positive correlation. Another study described the opposite for income alongside an inverse relationship between education levels and water conservation [24]. Moreover, some other studies reported that, in general, water conservation activities are normally associated with higher-income groups. Furthermore, the review also highlighted that individuals who are more educated, have smaller families, have smaller properties and own their own homes conserve more water than others [5]. To understand the dynamics of water use and how best water conservation problems can be solved, several factors can be considered such as income, water tariffs, family size, family upbringing, among others. However, this problem in context is site-specific; that is, the water use problem is unique to different locations and at different times. This is because different countries have their unique water problems and coping mechanisms in place to face water scarcity or reduce water use.

## 4. Conclusions

The results of the study indicated that the participants from rural households have a better water-saving attitude compared to the participants from urban households. This was a result of the scarcity of water resources in the rural community. This study shows that the availability of water does influence water use attitude. The urban community needs to learn and inculcate the water-saving culture from the rural community. This will reduce water use holistically in the country. It is, therefore, necessary for the government and other stakeholders to place a greater emphasis on decision implementation campaigns to encourage water conservation, especially in urban households. By understanding what drives attitude and what incentivises better water conservation practices, government officials can implement more suitable and targeted water conservation management interventions. It is also argued that while the results of this study may be true, water conservation remains context-specific and is dependent on some other underlying factors not mentioned in this study. For instance, higher-income households may consume more water than lower-income households with restricted water access.

Increasing water tariffs and bills, especially in the urban community, was a major determinant of the participants' water conservation attitudes. However, the rural com-

munity enjoyed free water supply from the community taps and did not need to worry about water bills. Knowing this, it could be ideal to point out that altering the attitudes of South Africans towards water conservation can be achieved on a micro scale based on an understanding of the problems associated with each specific community, especially issues on water bills. Furthermore, the study shows that variables such as family upbringing, the immediate environment and water scarcity had a positive influence on both communities' responses towards water use issues. However, advertisement did influence the water consumption of the participants in the urban community but had no significant influence on the participants in the rural community. The rural participants highlighted that they did not have enough advertisements about water scarcity.

Based on the findings of the study, it is recommended that the government should implement a policy on regulated water outlets, such as taps, showers and bathing outlets, with a standardised minimum water flow. Households should be encouraged to install greywater collection systems to reduce water wastage and improve water reuse. The government should also provide proper water management and distribution systems, especially in rural communities where piped water can reach every household. More workshops, advertisements and seminars should be introduced as a way of conducting educational and enlightenment campaigns regarding water scarcity, water wastage and greywater collection and reuse. The government can introduce a rationed allocation (shedding) of domestic water in urban communities to draw attention to the prevalence of water scarcity in the nation. Research should be conducted on cheap and easy purification and distribution of other sources of water, for example, seawater, due to the rapid loss of global freshwater. Water quality sustainable programs should be implemented in rural communities to combat high salinity and fluorides in water bodies. More of these programs should be put in place for the purification and sustenance of the nation's dams. The number of water outlets in households can also be reduced to save water.

**Author Contributions:** Conceptualisation: P.O.N. and O.S.D.; methodology: P.O.N., O.S.D., S.E.U., and R.M.; supervision: O.S.D., S.E.U. and R.M.; writing—original draft: P.O.N.; editing: O.S.D. and S.E.U.; writing of the final draft: P.O.N., S.E.U., O.S.D. and R.M. All authors have read and agreed to the published version of the manuscript.

**Funding:** This research received no external funding.

**Institutional Review Board Statement:** This study was approved by the Ethical Committee of the University of Venda (certificate number: SES/21/ERM/06/0306).

**Informed Consent Statement:** The authors state that they have obtained an appropriate institutional review board outlined in the Declaration of Helsinki for all human or animal experimental investigations. A signed informed consent document has been obtained from all participants included in the study.

**Data Availability Statement:** Available on request from the corresponding author.

**Acknowledgments:** The authors acknowledge with thanks the Directorate of Research and Innovations Committee of the University of Venda and the Centre for Postgraduate Studies, and the Postdoctoral Fellowship/Research Committee of the Cape Peninsula University of Technology.

**Conflicts of Interest:** The authors declare no conflict of interest.

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
