# Peer review of "Investigating the Attitude of Domestic Water Use in Urban and Rural Households in South Africa"

_water, doi:10.3390/w14020210_

Round 1

Reviewer 1 Report

Water use and the demand side policy responses to mitigate future domestic water variability will benefit from a deeper understanding of the current household water use, perception of water use, attitude towards water use and factors influencing household water use.The research has theoretical and practical significance.Suggestions are as follows:

  • The drawing in this paper is not standard, there is no scale, and it is not clear,such as Figure 1.
  • The table needs to be in three-line format.
  • In this paper, there is a lack of validation of the validity of the data, and more of them stay on the statistical description and lack of depth.
  • It is suggested to add a discussion section. Is there any reference for other regions and countries? In addition, what is the theoretical contribution of this research?

Reviewer 2 Report

Reviewer’s response

This review is about a manuscript entitled “INVESTIGATING THE ATTITUDE OF DOMESTIC WATER USE IN URBAN AND RURAL HOUSEHOLDS IN SOUTH AFRICA”. It makes a comparison of water use behavior between urban and rural households in two places of two provinces in South Africa. The topic is interesting as water scarcity is gradually affecting nations where water supply is usually sufficient. Introduction is nicely written but rest of the parts must be improved significantly. Methodology is not well described. There is a lot of confusions and lack of clarity throughout the section. ‘Reconnaissance survey’ is not common in this field of study which needs to be clearly described to be understood by the readers of diverse fields. Sampling strategy / sampling methods are not properly explained, and selection criteria is completely ignored. Results and discussions should be significantly improved. People in water scarce areas, all over the world, have their own coping strategies to sustain their lives with scanty water amount. In this respect, the results from rural households are not new to this field of study (see Pattanayak, S.K.; Yang, J.; Whittington, D.; Balkumar, K.C. Coping with unreliable public water supplied: Averting expenditure by households in Kathmandu, Nepal. Water Resour. Res. 2005, 41). The results of rural households should be explained as examples for urban households to improve their water saving attitudes and water use behaviors. Simple comparison between two types of households is less meaningful. In addition, the discussion should be done in a way that the paper has some scope among international communities. Currently, the paper only has scope within South Africa. Conclusions should be rewritten. Simple but troublesome mistake is that the manuscript neither has page number of line number which add difficulty to the reviewers. My specific comments are as below:

  1. Figure 1 title: Is Durban a part of Vheme district? Since Durban and Thohoyandau both are there, please provide an appropriate title. Improve figure resolution.
  2. “Formal homeland”: Add explanations.
  3. What percentage of households in rural area ‘Thohoyandau’ have piped water connection?
  4. RDP standard: What is this? Does it have some value/ figures?
  5. ‘35’ should be ‘35%’ (?) in the context of stolen water in Durban.
  6. Stage 1 should be detailed.
  7. Do you mean Thohoyandau by rural and Durban by urban? If yes, mention clearly.
  8. Wadoux et al.: Add simple description enough to understand what has been done by them. Do not let readers and reviewers to search, download, and read the whole paper by themselves.
  9. Raosoft software: What are parameters considered to calculate the sample size? What values you set for them? Please add simple description.
  10. Stage 2: Please clearly explain. Mention how the participants were selected, sampling methods. What were the selection criteria ?
  11. How many urban and how many rural households were selected?
  12. “The use of structured ……………..their perception.”: The sentence is unclear.
  13. Details on ‘water variable guides’ are unnecessary.
  14. Data analysis: Mentions which statistical tests were caried out (for example for table 2/3 results) and for what purpose the tests were carried out.
  15. Ethical approval: Did the authors took consent with the participants? Did the authors need to be approved by any ethical review board of their institution before conducting the study for including human participants?
  16. Results and discussion section: please go through general comments once again.
  17. ‘Wastewater’ is a term used for sewage. Instead of it you can use ‘used water’.
  18. ‘Chronic water users’: Please explain.
  19. ‘water saving’ and ‘wastage attitudes’ are different?
  20. 2 Stage 2 experiment: How do you conduct ‘demographic characteristics’? Should it be ‘demographic characteristics were analyzed’?
  21. Do you have participants of age 11 – 15 years as well? Can they respond relevantly? Can they understand questions and answer appropriately? Can their responses be considered as valid? What are the selection criteria of the participants?
  22. Table 2&3: What is the significance of the significance value? Mention the significant differences between two groups in text as well and explain how the difference can affect the answers and results. Plus, P-value = 0.000 should be represented as P-value < 0.0001.
  23. Family upbringing: You have accessed their ‘perception if family upbringing affects water use behavior or not’. You have asked if they ‘think’. So, it is the perception rather than ‘actual influence’. If you want to show influence of ‘family upbringing’ on participants’ water, use behavior, then more than one questions would be more suitable such as ‘do you use water profusely because you have seen your parents doing so’? I hope this difference is understandable.
  24. Table 4 & 5: 2X2 format is not needed because the correlation is only between two variables. You can present the modified version of the table that SPSS provided.
  25. Please explain how the correlation might have observed. What could be the reasons to obtain these results?
  26. Last paragraph of ‘Results and discussion’ section: What is your view in relation with your own findings?
  27. Conclusion: Re-write. First summarize your findings and then discuss which result can be useful to derive what recommendation.

Reviewer 3 Report

I think this is a very meaningful study on water demand management in a water-stressed country. However, there are some problems with the statistical analysis, the way the sampling and the insufficient discussion. Therefore, I think it cannot be published as it is.

English letters related to statistics (ex. p) should be italicized.

Abstract

What is (-0.306)?

Page 2

What is “desalination of wastewater treatment plant”?

Page 3

“4. Are the people conscious of the amount of actual water used and do their perceptions of water use correspond with actual water use?”

I cannot understand how to confirm this point.

Figure 1

The text in the map is hard to read.

Figure 2

The legend (explanatory note) is difficult to read. Some of the pie charts are broken. Why are only Vhembe's graphs shown in detail?

Page 5 Stage 2

How to select 300 participants?

Table 2 and 3

Please think significant digit.

What is the significance? What king of statistical analysis did you use?

Page 10 3.5 Immediate environment

Authors said in the last part that immediate environment is related to the difference in water use between urban and rural. But I think that is not true. Isn't it a difference in the availability of water?

Page 10 3.6

It is better to include in 3.1 to reinforce your opinion.

Page 11

The answer to the water saver is not equal to water use.

Page 12

“Furthermore, the review also highlighted those individuals who were more educated, had

smaller families, smaller properties and owned their own homes conserve more water than others.” It is review based statement. How is the situation of this study?

“For instance, higher income households may consume more water than lower-income households with restricted water access. Also, urban households may conserve more water than rural community, because higher tariffs bills for the urban household.”

I cannot understand why you can say so.

“Based on……”

How authors came to this conclusion? I have no idea from the study results.

Round 2

Reviewer 1 Report

I'm so glad to see that the author made careful revisions in accordance with the revised comments. The paper has reached the publication requirements.

Author Response

Thank you for your wonderful contribution toward the success of this manuscript. 

Reviewer 2 Report

Thanks for making the revisions. However, it is very much difficult to follow where you have made changes. Please provide line numbers in the manuscript and mention them in response sheet as well. So that it will be easy for reviewers to quickly reach the revisions and evaluate them. I had mentioned about giving line numbers in general comments as well. But it seems to be ignored completely. In addition, I checked the authors' response sheet but few comments are unanswered. So, please respond them as well. Hence, I have attached previous comments as it is. 

Author Response

Thanks for giving us the opportunity to submit a revised version of our manuscript. We believe we have responded to the reviewer’s comments to the best of our ability and thanks to them for giving useful comments to improve our manuscript. Number lines of our responses to each comment have been incorporated for easy verification.

Please view the manuscript below with the number lines

Thanks

Round 3

Reviewer 2 Report

  1. The discussion should be done in a way that the paper has some scope among international communities. Currently, the paper only has scope within South Africa.
  2. Definition of 'reconnaissance survey' will help readers to understand the method.
  3. L149-L154: Explanation on the software is unnecessary. Enlist the parameters entered to produce sample size.
  4. On what the random sampling technique was applied? Was it whole name list of the population? 
  5. Table 2&3: What is the significance of the significance value? If some relationship showed significant association then what does it implies. You have just reported results. Please add what does that result mean or imply?

Author Response

Thanks for giving us the opportunity to submit a revised version of our manuscript. We believe we have responded to the reviewer’s comments to the best of our ability and thanks to them for giving useful comments to improve our manuscript. Number lines of our responses to each comment have incorporated for easy verification.

  1. The discussion should be done in a way that the paper has some scope among international communities. Currently, the paper only has scope within South Africa.

The discussion has been highlighted in yellow in the manuscript

  1. Definition of 'reconnaissance survey' will help readers to understand the method.

The definition of 'reconnaissance survey' has been added in the manuscript (line 150 – 153).

  1. L149-L154: Explanation on the software is unnecessary. Enlist the parameters entered to produce sample size.

The parameters have been added and defined in the manuscript (Line 156 – 16).

  1. On what random sampling technique was applied? Was it whole name list of the population?

Random sampling was applied to the sample size (n) and not the entire population (141 - 143).

  1. Table 2&3: What is the significance of the significance value? If some relationship showed significant association then what does it implies. You have just reported results. Please add what does that result mean or imply?.

The implication for the significant association has been added (Line 288 – 290; 301 – 305; 330-332; 344-346; 355-356; 378-380; 388-390).
